# Can Pay-As-You-Go, Digitally Enabled Business Models Support Sustainability Transformations in Developing Countries? Outstanding Questions and a Theoretical Basis for Future Research

**David Ockwell** [1],*, **Joanes Atela** [2], **Kennedy Mbeva** [2], **Victoria Chengo** [2], **Rob Byrne** [3], **Rachael Durrant** [3], **Victoria Kasprowicz** [3] and **Adrian Ely** [3]

[1]  Department of Geography and ESRC STEPS Centre, School of Global Studies, University of Sussex, Brighton BN1 9SJ, UK

[2]  African Centre for Technology Studies (ACTS), ICIPE Duduville Campus, Kasarani, P.O. Box 45917–00100, Nairobi, Kenya; J.Atela@acts-net.org (J.A.); K.Mbeva@acts-net.org (K.M.); V.Chengo@acts-net.org (V.C.)

[3]  SPRU (Science Policy Research Unit), Sussex Business School, University of Sussex, Brighton BN1 9SL, UK; R.P.Byrne@sussex.ac.uk (R.B.); R.Durrant@sussex.ac.uk (R.D.); v.kasprowicz@sussex.ac.uk (V.K.); a.v.ely@sussex.ac.uk (A.E.)

*  Correspondence: d.g.ockwell@sussex.ac.uk

**Abstract:** This paper examines the rapidly emerging and rapidly changing phenomenon of pay-as-you-go (PAYG), digitally enabled business models, which have had significant early success in providing poor people with access to technologies relevant to the Sustainable Development Goals (SDGs) (e.g., for electricity access, water and sanitation, and agricultural irrigation). Data are analysed based on literature review, two stakeholder workshops (or "transformation labs"), and stakeholder interviews (engaging 41 stakeholders in total). This demonstrates the existing literature on PAYG is patchy at best, with no comprehensive or longitudinal, and very little theoretically grounded, research to date. The paper contributes to existing research on PAYG, and sustainability transformations more broadly, in two key ways. Firstly, it articulates a range of questions that remain to be answered in order to understand and deliver against the current and potential contribution of PAYG in affecting sustainability transformations (the latter we define as achieving environmental sustainability and social justice). These questions focus at three levels: national contexts for fostering innovation and technology uptake, the daily lives of poor and marginalised women and men, and global political economies and value accumulation. Secondly, the paper articulates three areas of theory (based on emerging critical social science research on sustainable energy access) that have potential to support future research that might answer these questions, namely: socio-technical innovation system-building, social practice, and global political economy and value chain analysis. Whilst recognising existing tensions between these three areas of theory, we argue that rapid sustainability transformations demand a level of epistemic pragmatism. Such pragmatism, we argue, can be achieved by situating research using any of the above areas of theory within the broader context of Leach et al.'s (2010) Pathways Approach. This allows for exactly the kind of interdisciplinary approach, based on a commitment to pluralism and the co-production of knowledge, and firmly rooted commitment to environmental sustainability and social justice that the SDGs demand.

**Keywords:** Pay-As-You-Go; digitally enabled business models; sustainability transformations; energy access; energy for development; sustainable development goals

## 1. Introduction

Sustainable Development Goal (SDG) 7 and accompanying Agenda 2030 goals aim to provide access to sustainable energy for all (SE4ALL) by 2030. This recognises the fundamental need for access to energy for improved economic productivity in many developing countries where energy consumption is tightly coupled to economic growth [1]. It also recognises the human and economic development needs of many poor and marginalised women and men, 1.1 billion of whom currently lack access to electricity [2]. SDG7 also recognises the tension with achieving such goals within a carbon-constrained world. It looks to leverage potential synergies with new, sustainable energy technologies and off-grid energy access in rural, urban, and peri-urban contexts. A key problem to date has been finding ways of making access to sustainable energy technologies affordable in contexts characterised by extreme poverty [3]. This issue applies equally to many other SDG-relevant technologies (e.g., technologies for improving agricultural productivity, water pumping, and health), as affordable access to new technologies and innovation cuts across multiple SDGs (as does, often, the need for access to electricity).

No matter how one looks at these problems, a goal such as SE4ALL inevitably requires transformative changes at speeds that are, to date, unprecedented. Current analysis, however, suggests we are not on track to achieve the UN's SE4ALL goal [2]. It is no wonder, then, that against this backdrop the recent emergence and rapid proliferation of new, digitally enabled Pay-As-You-Go (PAYG) business models (now colloquially referred to as "PAY-Go" by practitioners), that aim to make access to sustainable technologies affordable to poor people, has captured widespread international donor and press attention. For example, the World Resources Institute (WRI) describes them as a "one-stop-shop solution" to the SE4ALL challenge [4] and *The Economist* describes them as "ending energy poverty" by providing "power to the powerless" [5]. Even detailed, financially grounded market analyses tend to talk up the promise of PAYG. For example, the global trade association for mobile operators describes PAYG as the "perfect example" of the "second wave of inclusive digital innovation" [6], p. 4.

PAYG business models use an innovative combination of mobile banking technologies and machine-to-machine (M2M) technology to facilitate PAYG payment plans that seek to overcome barriers experienced by past micro-finance models, such as high up-front initial costs and lack of integration and availability of technical support [7]. In many instances, these payments either mirror, or are cheaper than, the existing payments customers make for alternative energy sources (e.g., kerosene for lighting and cooking). The machine-to-machine technology also enables providers to switch off the system if customers fail to top-up their payment plans, or if they default (different companies have different definitions of what constitutes "default"—e.g., the anonymous company studied by Barrie and Cruickshank [8] considered customers to have defaulted if they had not added credit to their systems for a four-week period). Through this they are providing access to sustainable electricity and other technologies and services (e.g., energy efficient appliances, agricultural irrigation technologies, and water and sanitation services [6]) to customers for whom access to such technologies and the services they provide would otherwise be prohibitively expensive.

Despite their promise, however, PAYG business models are very new, and very little critical analysis of their actual and potential implications for delivering against the SDGs has been conducted. As the literature review in Section 2 demonstrates, to date, only five peer-reviewed papers in mainstream academic journals exist on PAYG business models, as well as seven papers in peer-reviewed practitioner journals and six detailed professional reports. Having first emerged in Kenya circa 2010 following the widespread success of the MPESA mobile banking system, by 2016 a UC Berkeley study by Alstone et al. reported 30 PAYG companies operating in 32 countries in the Global South [9]. Alstone et al. [9] do not state specifically which countries these are, but a map provided in the report suggests that they include coverage in Central and South America; East, West, and Southern Africa; and South and Southeast Asia. The Groupe Spéciale Mobile Association (GSMA) [6] reports the majority of the market share being focussed in East Africa (Kenya, Tanzania, Uganda, and Rwanda),

with evolving markets present in West Africa (Ghana and Ivory Coast) and South Asia (India). Another report by the GSMA, published just eleven months later, listed many more countries as having active PAYG-based companies [10]. Whatever the actual numbers, the differences in this reporting imply the spatial distribution of PAYG activity is evolving rapidly.

As well as this rapid geographical spread, stakeholder interviews and workshops conducted in support of our analysis below also suggest a proliferation of different types of PAYG business models. Some, for example, provide only PAYG software-based platforms for other companies to use (in whatever application those companies want—from electricity supply and household appliances, to agricultural irrigation and transport), whilst others provide whole-system technology and finance services, from purchase, through maintenance, to end-of-life and recycling. This diversity is also reflected in the PAYG business model being used in multiple other SDG-relevant sectors beyond electricity access, where PAYG first began, but with a continued emphasis on providing access to services (e.g., water, sanitation, cooking, and irrigation) to poor and marginalised people in rural areas [10,11], thus emphasising the potential relevance of the PAYG phenomenon across multiple SDGs. As Alstone et al. [9], p.10, put it:

> " ... the PAYG market is incredibly dynamic. New business models appear almost daily, companies pivot and change approaches, funding is raised, and players disappear. This report presents a snapshot of what our team observed during the June and July of 2014. Since then, even our partners have altered their approaches ... "

Against this backdrop of a dearth of critical, peer-reviewed scholarship and the rapidly expanding and metamorphosing shape of the PAYG space in developing countries, we seek in this paper to make two key contributions. Firstly, based on consultation with stakeholders in Kenya, the paper articulates a number of critical questions pertaining to the current and potential contribution of PAYG business models for delivering sustainability benefits. Our aim here is to highlight key dimensions to the interaction between PAYG business models and sustainability that are not acknowledged within existing mainstream discourse. Stakeholder consultation is based on seventeen interviews and two stakeholder workshops (or, more specifically, two "transformation labs"—see the methodology below) with stakeholders from Kenya and elsewhere in East Africa, where PAYG business models first evolved, and where most experience to date has been accrued. The process of stakeholder selection and the rationale for different stakeholders' involvement is described in the methodology below.

In line with Leach et al. [12], we define sustainability as foregrounding concerns with both environmental sustainability and social justice/inclusivity. Our particular interest is in the intersection between PAYG business models and the myriad ways in which access to energy—and technology and innovation more broadly—are interwoven across the majority of the SDGs. In aligning ourselves with Leach et al.'s work, and in particular their Pathways Approach to analysing sustainability, we adopt the position that any existing or potential 'benefits' accrued from PAYG are contingent (both in their definition and their perceived receipt) on the self-constructed needs and aspirations of different stakeholders. We further maintain that these self-constructed needs and aspirations will differ between different actors, for example—benefits perceived by an economically marginalised woman in a remote rural area will be different from benefits perceived by a policy maker in an intergovernmental development organisation (see Ockwell and Byrne [13] for an application of a Pathways Approach perspective to solar photovoltaics (PV) for energy access in Kenya). By adopting a methodology that explicitly focusses on eliciting stakeholder perspectives and operationalizing a normative commitment to the co-construction of knowledge, we endeavour to do justice to understanding and responding to the self-constructed nature of 'sustainability benefits'.

Having articulated these critical questions regarding the sustainability benefits of PAYG, the paper then seeks to make a second contribution by articulating four areas of conceptual work within the emerging critical social science literature on energy access, and sustainability transformations more broadly, that may be of use in facilitating future, critical research on PAYG and sustainability transformations. Our intention is to facilitate future research focussed on fulfilling two key aims:

1) understanding in further detail the sustainability benefits (and potential drawbacks) of PAYG, and 2) understanding the opportunities and barriers to increasing the contribution of PAYG to sustainability in the future. At all times, in line with our definition and position on what constitutes sustainability articulated above, we foreground attention to issues of inclusivity, with a consistent focus on who gains, who loses, and how and why, and how this might change in ways that are more inclusive in the future (cf. [14–16]). In this way we are fundamentally interested in how PAYG business models might contribute to rapid transitions, or 'transformations', in ways that foreground issues of environmental sustainability and social justice and do not underplay the deeply socio-cultural and political nature of transformation (cf. [17]).

The paper begins with a review of the small existing peer-reviewed and grey literature on PAYG business models before outlining the methodology that underpinned the new data collected for this paper, which are analysed in the subsequent section. The paper then suggests three key levels of analysis and linked areas of relevant theory that constitute a potential basis for progressing future, critical research in this field. A fourth area of theory is also suggested as a means of providing an overarching framework, through which research utilising the initial three areas of theory might be coordinated within a normative commitment to pluralism and the co-construction of knowledge.

To be clear from the outset, we are not claiming that this paper provides a comprehensive answer on the current and potential future impacts of PAYG for sustainability transformations. Nor are we claiming that the conceptual approaches we propose below are the only ways in which future research can, or should, be framed. Our hope is, rather, that the ideas and insights herein provide a nuanced basis for critical, but constructive, future research that is grounded in the existing literature on PAYG, the emerging critical, socio-cultural turn in energy access research more broadly [13], and in the perspectives of different stakeholders. In line with many of the stakeholders and commentators active in the PAYG space, we share a normative commitment to delivering against the SDGs and to a vision of sustainability that is both environmentally sound and socially just. Our hope is that this paper can, therefore, make a small contribution to future research that contributes to realising these commitments.

## 2. Pay-As-You-GO (PAYG) Business Models: Insights from Analyses to Date

### 2.1. Coverage of Existing Literature

In this section, we summarise the peer-reviewed and grey literature available on PAYG business models to date. As inferred above, the existing literature is at best patchy and a long way from providing any comprehensive overview of what is a rapidly expanding and metamorphosing phenomenon. A literature search using Scopus, Web of Science, Science Direct, and Google Scholar, together with direct approaches to authors of some of the papers these searches identified to ask if they were aware of other literature we had missed, resulted in identification of only five articles in mainstream academic journals [8,9,12,18,19], seven articles in practitioner journals [20–26], and six detailed reports [5,7,10,11,27,28]. Of these latter, three were produced by the global mobile operators' key trade association (the GSMA) [7,11,27], one by researchers from UC Berkeley based on a study commissioned by the World Bank [10], one by the World Resources Institute [5], and one by an independent think tank, the Consultative Group to Assist the Poor [28]. Whilst not explicitly stated, it is likely that only the UC Berkeley and World Resources Institute reports were subject to peer review (and probably not blind peer review).

Within this small literature, only three papers attempt to use any kind of theoretical approach in their analysis. Rolffs et al. [8] operationalize a "strategic niche management" perspective from the socio-technical transitions literature. Barrie and Cruickshank [9] operationalize a "diffusion of innovations" theoretical perspective, which originates from the work of rural sociologist Rogers [29]. Finally, Bisaga and Parikh [18] speak to the theory of the "energy ladder" from energy access/energy and development research. Bearing in mind that within the social sciences it is theory, built from multiple observations across myriad different contexts, that supposedly allows research to work

towards more generalizable insights. This lack of theoretical engagement within the literature to date calls into question the extent to which insights from research on PAYG are in any way generalizable.

Moreover, the literature to date, including the current paper, is based largely on data from East Africa, with very little on PAYG applications elsewhere in the Global South. There have also been no longitudinal studies conducted to date, making it difficult to comment on the temporal sustainability of any PAYG business models and their attendant costs and benefits, whether financial, social or environmental. As Alstone et al. [10], p. 7, put it:

> "The strength of connected PAYG (and other connectivity-enabled approaches) for supporting reliable, adaptive solar energy access will become clearer as the first wave of systems entering the market today age and are supported with maintenance, expansion, and replacement."

One other notable characteristic of this emerging literature is the use, in three cases [9,12,18], of remote, mobile-enabled data, collated by PAYG companies, on consumer practices, consumer financial risk profiles, and the technological performance of solar home systems and other technologies to which PAYG businesses facilitate access (e.g., battery performance, charging profiles, and impacts of weather). In the case of the three aforementioned peer-reviewed analyses, these data have facilitated sophisticated analysis that is able to look in detail at the practices of consumers. Some commentators also note that PAYG businesses themselves are using the remote data to which PAYG affords access in order to generate consumer debt-financing risk profiles, as well as discussing the potential need for developing more centralised risk profiles, available to all actors across the sector, in efforts to facilitate the scaling up of PAYG-based services [7,11]. Whilst access to such data does facilitate interesting and insightful analysis (e.g., see our discussion of social practices later in this section), there are significant potential ethical issues related to the use of such data, issues that some authors acknowledge [12] and others note as having arisen as salient concerns within focus groups with PAYG users [10].

## 2.2. Dominant Focus on Technical and Financial Dimensions

As with the majority of the energy and development literature to date, there is a tendency within what literature so far exists on PAYG to privilege a focus on technical and financial issues, with relatively little, or at best implicit, attendance to socio-cultural or broader innovation system-related dimensions. Other than passing mention of "enabling environments" regarding the uncertainty around relief of value-added tax on solar imports [8,24], or a couple of mentions of the word "politics" without any elaboration [8], consideration of the political or political economy dimensions of PAYG's contribution to sustainable development is completely absent. This is not to diminish in any way the importance of financial and technical dimensions. Rather it is to highlight, as we expand upon further below, the limitations of such a two-dimensional perspective in underpinning anything close to what might be considered transformative change. Indeed, Rolffs et al.'s [8] core thesis is that, despite PAYG business models being ostensibly financial and technical in nature, it is actually the deep understanding of the social practices of poor and marginalised people in paying for and consuming energy services—which was developed by early pioneers of PAYG through decades of living and working in poor communities—that is fundamental to the early success and potential future growth of PAYG. It is through such understandings, they argue, that these pioneers were able to identify the opportunity to use mobile payments, machine-to-machine technologies, and increasingly cheaper solar PV to construct PAYG payment plans that come close to replicating existing payments for energy (e.g., kerosene) practices around purchasing mobile air time, or using mobile banking technologies.

Within the analysis of technical and financial dimensions in the existing literature, four issues in particular tend to be emphasised. Firstly, several authors emphasise the benefits of PAYG over traditional micro-finance-based approaches [8,22,23], particularly when targeting lower income households with access to basic energy services [19]. Secondly, the availability of mobile banking services is discussed by several authors as having implications for the viability of PAYG. Whilst

attempts have been made by some businesses to overcome this—for example, via the use of scratch cards—these have been plagued by logistical limitations relating to making cards available in remote locations and gathering cash from vendors [24]. Zollmann and Waldron [28] note experimentation with other revenue collection approaches, including cash collection and exchange of mobile airtime for solar credit, but find that such approaches suffer from significantly higher cost and coordination issues than mobile banking-based PAYG business models.

A third issue emphasised in the existing literature refers to the difficulties companies face in raising working capital [5,10,19,26,28]. This is due to a range of issues, including the need for long repayment terms and foreign currency risks, a lack of data on customer repayment ability/credit ratings, high transaction costs associated with mobile money, and a lack of experience and understanding of PAYG business models by commercial banks within the countries of operation. The latter has been hypothesised as one reason why most PAYG companies are foreign owned and headquartered [5] (an important issue in relation to questions of how much local companies and countries are benefiting from PAYG, which comes up in the stakeholder consultation and political economy discussion further below). In light of this, it is not surprising that many PAYG companies have originally relied on donor and grant funding for their initial set up [21,24,25]. There is likely still a key role for public funding for PAYG businesses, with a strong public good rationale for such investment in providing access to SDG-relevant technologies and services (although, learning from Bangladesh, Sanyal and Prins [5] emphasise the need to channel such funding in local currency through local banks to increase national understandings of PAYG).

The final area of financial and technical issues reported in the literature concerns the high costs and complexities of developing and implementing technology interfaces that can deal with myriad different potential network operator protocols (protocols that are sometimes subject to change [24]), different mobile money platforms, customer interfaces, etc. [10,12,28]. This is compounded by low levels of digital literacy amongst some consumers that demands simple consumer interfaces [28]. These costs can act as market barriers to small operators [12]. Some authors also report issues around highly technical choices regarding the suitability of GSM connectivity, global vs. local networks, embedded SIM options, and so on. This further emphasises the levels of technical capabilities companies require to work in this space [7].

### 2.3. Socio-Cultural Dimensions of PAYG

Socio-cultural dimensions of the PAYG phenomenon are largely ignored in the literature to date, which is, as developed further below, both concerning and noteworthy given that the first paper on the subject in a mainstream academic journal (i.e., [8]) argued that it was an understanding of socio-cultural practices that was fundamental to the early success of PAYG. That said, the existing literature does touch on several relevant issues. The first regards the extent to which PAYG reaches the poorest consumers. Muchunku and Kirsten [19], for example, report some vendors being paid commission based on levels of consumer default, providing a disincentive for vendors to target lower income people. Collings and Munyehirwe [24] also report consumers being of above average wealth within rural populations. That said, Alstone et al. [10] argue that PAYG still represents the best finance model to date for reaching lower income, especially rural, populations because of its potential to lower the transaction costs of energy access.

One key socio-cultural dimension that is almost completely ignored in the literature to date is that of gender (and any other characteristics of potential social exclusion). Nowhere is there any explicit, gender-focussed research on the ways in which PAYG might impact positively or negatively on gender relations (an issue discussed in more detail further below). Barrie and Cruickshank [9] do mention gender, noting that their respondents reported women being in charge of payments, leading (they report) to men feeling justified in wanting to keep a solar home system despite payment-defaulting. Zollmann et al. [28] also touch on gender, reporting purchase decisions for PAYG as being made by men despite resistance from their wives whose household budgets are often negatively impacted by

the purchase (including when decisions are made to acquire more assets once the initial solar loan is paid off).

At multiple points in their detailed analysis, Barrie and Cruickshank [9] mention various other socio-cultural issues. For example, cultural clashes between ideas of possession vs. ownership were observed to make repossession of PAYG equipment in the event of default difficult, especially where customers were accustomed to free handouts from past development projects. Fluctuations in income and ability to top up payment plans due to the nature of customers' livelihoods were also noted (e.g., agricultural incomes, or highly-mobile market traders—the former also being noted by Bisaga and Parikh [18]). This inflexibility towards seasonal incomes potentially undermines Rolffs et al.'s [8] point on the close match between PAYG payment plans and existing consumer practices around paying for energy. But the idea of increased potential for default is countered by Bisaga et al.'s [12] analysis, where they cite the ability of machine-to-machine technology to switch off services as serving to decrease possibilities of default. It is not clear whether the business analysed by Barrie and Cruickshank [9] had this capability or not, but it is notable that Bisaga et al.'s [12] sample size was significantly larger than Barrie and Cruickshank's [9] (the former basing their analysis on 20,000 active systems vs. the latter's 747).

Elsewhere in the literature, Bisaga and Parikh [18] explicitly refer to social practice theorists when reporting on observed shifts in social practices once PAYG systems are installed, where consumers are observed to change their practices to mirror the availability of energy in a solar home system. They also note the relevance of the availability of appliances in influencing this. Collings and Munyehirwe's [24] analysis also demonstrates implicit recognition of the relevance of social practices through their emphasis on the similarity between scratch cards for solar and for air time as being a factor that increases ease of use for customers.

Finally, it is worth noting that the "diffusion of innovations" theory applied by Barrie and Cruickshank [9] does include attending to a notion of the "social". One example of this is their inclusion of the "nature of the social system" as a key analytical variable. But this is then rather crudely described as social systems being either "urban" or "rural", with the implicit assumption of a lack of variability between different socio-cultural attributes within different rural or urban locales. They also acknowledge in their discussion that new innovations being different to existing "practices" can be a key factor mitigating against widespread adoption of a new technology. It is interesting, however, that it is the solar home system that they note as matching existing consumer practices, and not, as in the case of Rolffs et al. [8], the PAYG payment structure.

*2.4. Innovation Systems*

The final dimension that is acknowledged as important elsewhere in the energy and development literature, but largely ignored in the existing PAYG literature, is innovation systems (e.g., see [30]). Barrie and Cruickshank's [9] analysis does touch on aspects that other authors (e.g., [13]) emphasise as important parts of building functioning innovation systems to increase uptake of new, SDG-relevant technologies. These include the use of market research, proactive demonstration, promotions, and marketing; although, the initial boost in sales in Barrie and Cruickshank's [9] study that resulted from such activities was then reflected in subsequent higher default rates, thus questioning the sustainability of such approaches. Their analysis is not, however, explicitly concerned with a systemic perspective on innovation and development, or its socio-technical nature, which we expand upon in Section 5 below.

Nowhere in the existing literature on PAYG, then, do we see any comprehensive, theoretically grounded analysis of the PAYG sector, even at a single country level (not surprising given the rate at which it is emerging and its increasing diversity), or any longitudinal analysis beyond very short periods of time. In line with the insights from the stakeholder consultation undertaken for this paper (see below), there is clearly a range of critical questions that remain unanswered, and which are vital for research to engage, if the potential for PAYG to contribute towards rapid sustainability transformations is to be properly understood and realised in practice. Having set out the key insights from the PAYG

literature to date, we now turn to the empirical data collated from stakeholder consultation for the current paper, before setting out some of the key areas of theory that might help to frame future research aimed at better understanding and supporting the existing and potential role of PAYG in underpinning sustainability transformations.

## 3. Methodology

As the literature review above reveals, the current state of knowledge on the potential for PAYG to facilitate sustainability transformations is currently in its infancy. It is at best geographically patchy, based on a relatively small number of studies over relatively short time-scales. That said, there are clearly several interesting hypotheses beginning to emerge from this literature, such as regarding the nature of innovation, that might serve the needs of low-income men and women and the need to understand existing social practices around consuming and paying for energy services, in order to develop effective technical and financial solutions to the energy access issue (responding to SDG7 and the myriad other SDGs which rely on access to clean, reliable energy, or on innovation and technology access more broadly).

Bearing this in mind, the project upon which this paper is based set out to consult with a cross-section of stakeholders with knowledge and experience of PAYG business models. The aim of the consultation was to elicit key questions and concerns that remain unanswered regarding:

1.　The existing and potential contribution of PAYG to sustainability transformations;
2.　Existing and potential barriers to rapidly scaling up PAYG business models;
3.　Which socio-technical innovations and governance approaches might hold the key to transforming the PAYG space in ways that lead to more sustainable and inclusive future benefits for all.

Stakeholder consultation consisted of two different methods, namely transformation labs, or "T-Labs", and semi-structured interviews, each conducted in one of three different phases of data collection. Based on earlier work on "social innovation labs" [31], T-Labs " . . . are specifically designed to guide transformations in social-ecological systems towards sustainability . . . [by] include[ing] a set of stakeholders who may have different roles and perspectives, but who have an interest in solving the problem and some ability to provoke change" [32], p.7. For a full explanation of the background and processes involved in T-Labs, readers are encouraged to consult the practical guide produced by the Pathways Network [32]. In summary, the process involves convening two, subsequent workshops (or "labs"), which move from an initial process of attempting to frame challenges and identifying potential innovators/stakeholders towards developing change strategies and even testing prototypes of interventions. "Prototypes could be new business models, services, or kinds of governance that fundamentally change human-environmental interactions and contribute to changes for a better future" [32], p.7. Importantly, the nature of T-Labs and the ways in which the processes evolve is contingent upon the context specificities of any given issue, whether these contexts pertain to social, political, economic, technical, or environmental considerations. Their emphasis on co-production of knowledge, via convening a plurality of perspectives and voices, is designed specifically to recognize and respond to such specificities (and thus aligns T-Labs with this paper's normative commitment to recognising a plurality of voices and the co-construction of knowledge as necessary for understanding the nature of sustainability transformations).

The type and number of stakeholders that participated in the two T-Labs are summarized in Tables 1 and 2 (note that private sector participants were all PAYG-based businesses, other than one participant who was a lawyer previously involved in drafting relevant legislation).

**Table 1.** Stakeholders participating in T-Lab 1.

| Type of Stakeholder | Number of Interviewees |
|---|---|
| Private sector | 6 |
| Public sector | 3 |
| Not-for-profit/third sector | 2 |
| Research/universities | 3 |
| Pay-as-you-go (PAYG) users | 1 |
| Total | 15 |

**Table 2.** Stakeholders participating in T-Lab 2.

| Type of Stakeholder | Number of Interviewees |
|---|---|
| Private sector | 3 |
| Public sector | 1 |
| Not-for-profit/third sector | 1 |
| Research/universities | 9 |
| PAYG users | 0 |
| Total | 15 |

T-Lab 1 focussed on framing existing problems and unanswered questions regarding the existing and potential contribution of PAYG to sustainability transformations (questions 1 and 2 above). T-Lab 2 focussed on articulating questions and issues that needed to be addressed in order to develop solutions to the issues raised in T-Lab 1 (question 3 above).

In between T-Labs 1 and 2, several stakeholders, including some of those involved in T-Lab 1 and other stakeholders contacted via a snowballing approach, were interviewed. Interviews were conducted in person and by Skype/phone and were semi-structured. The structure was based on asking the initial opening question of how they perceived PAYG to be contributing to access to sustainable technologies and the barriers to scaling up this contribution, then using the results of T-Lab 1 to drill down in more detail on the specific issues that had arisen. Additional questions were asked regarding potential solutions and possible future innovations to increase the contribution of PAYG to sustainable development, with a view to helping with the framing of the subsequent, solutions-oriented T-Lab 2. Stakeholders participating in T-Lab 1 and the interviews were also asked who they perceived to be key change-makers in the PAYG space in order to inform which stakeholders were approached to participate in T-Lab 2. The number and type of stakeholders interviewed is summarised in Table 3. Note that all, apart from one private sector interviewee, were companies operating around a PAYG business model. In the case of the interviewees, however, as opposed to the T-Labs, the one private sector exception was a UK-based telecoms solution company with experience in mobile network protocols, and they had specialist knowledge and experience with mobile applications in humanitarian contexts.

**Table 3.** Stakeholders interviewed.

| Type of Stakeholder | Number of Interviewees |
|---|---|
| Private sector | 10 |
| Public sector | 5 |
| Not-for-profit/third sector | 1 |
| Research/universities | 1 |
| PAYG users | 0 |
| Total | 17 |

Four of the participants in T-Lab 1 also took part in T-Lab 2, and one participant in T-Lab 2 was also interviewed. This brought the total number and type of stakeholder consulted through the research to 41 (as summarised in Table 4).

**Table 4.** Total number and type of stakeholders consulted through the project, including via the two T-Labs and seventeen interviews.

| Type of Stakeholder | Number of Interviewees |
|---|---|
| Private sector | 19 |
| Public sector | 8 |
| Not-for-profit/third sector | 3 |
| Research/universities | 10 |
| PAYG users | 1 |
| Total | 41 |

The results of the two T-Labs and the interviews were analysed to identify common themes and group different questions, concerns, and insights according to these themes (see Section 4). These themes were then compared to emerging themes in the nascent, critical social science literature on energy and sustainable development (see Ockwell and Byrne [30] for a recent review of this literature) to identify any concepts and theories that could usefully provide the basis for future, critical research in this field, with a firm focus on yielding an empirical analysis that is able to inform policy and practice in ways that will increase the contribution of PAYG to sustainability transformations (see Section 5).

## 4. Results: Critical Questions Arising from Stakeholder Consultation Regarding the Current and Potential Contribution of PAYG Business Models to Sustainable Development

A key insight to emerge from T-Lab 1 was that the problem space in Kenya and beyond around PAYG was closely associated with diverging interests, views, knowledge, and narratives on how to go about leveraging innovations like PAYG to underpin transformative change. T-Lab 1, together with subsequent stakeholder consultation (through the interviews and T-Lab 2), yielded multiple questions, which stakeholders viewed as currently unanswered, but critical to any future reliance on PAYG to achieve sustainability transformations. The questions arising from both T-Labs and the interviews can be grouped under the following four overarching themes:

1. **How do national contexts enable or constrain PAYG business models?**

Subquestions under this broader question included: Have the nature and strength of existing national stakeholder networks around key technologies, like solar PV and mobile banking, had significant impacts on the evolution and success of PAYG (and hence been material to the relative success and potential of these business models across different national contexts)? What aspects of national contexts do PAYG businesses perceive as constraining or enabling? What technologies and sectors are PAYG models currently being used for beyond energy? Is there potential for expansion across other technologies and sectors? How would all of this change if new ideas around open source software platforms for PAYG were developed? How would this impact existing PAYG actors? What role do mobile networks play? Can single mobile network dominance vs. multi-network diversity (as observed to be varying across different East African contexts) explain the relative success of PAYG in Safaricom-dominated Kenya, and is this material to potential for scaling up elsewhere? How do mobile network protocols enable and constrain PAYG, and what needs to change for PAYG models to scale up internationally?

2. **In what ways (positive and negative) do PAYG business models intersect with the lived realities of poor and marginalised women and men?**

Subquestions under this broader question included: Who currently benefits from PAYG (e.g., is it the poorest people, women, men, or other categories where social discrimination occurs)? And under

what conditions are benefits realised in practice (including different local contexts, different scales and types of technologies, and different structures and foci of different emerging PAYG business models)? Are financial, productive opportunities being enabled for users; if so, how, where, and for whom?

**3. What is the global political economy of PAYG business models?**

Subquestions under this broader question included: Where are benefits being accrued, by whom, how, and why? Are indigenous firms, research organisations, governments, and other actors gaining or losing? Where is capacity being built and capital being accumulated; is it within countries where PAYG businesses are operating or elsewhere (particularly bearing in mind most existing companies are headquartered in the Global North)? How is this influenced by dominant development discourses, particularly neoliberal development thinking and the increased emphasis on entrepreneurship and financialisation, and what is the implication of this for the agency of poor countries and women and men therein? What does this imply for how benefits of PAYG could be more firmly focussed on the countries where these business models operate?

**4. What do the answers to questions 1–3 imply about the potential for PAYG business models to deliver sustainability transformations and how do they contribute to theory building and future research on this issue?**

Subquestions under this broader question included: Under what circumstances do which actors gain and lose, how, and why? What are the broader implications of this for thinking about which sustainable development pathways are implied by which existing or possible future PAYG business models?

**5. Discussion: Theoretical Pathways for Future Research and Analysis**

The questions above focus our attention on three distinct levels of analysis with which PAYG business models intersect:

1. National contexts for fostering innovation, including technology development and uptake.
2. The daily lives of poor and marginalised women and men.
3. Global value chains and political economies.

Based on these three levels, and the broader concern of this paper to consider PAYG within the context of sustainability transformations, we have identified three distinct areas of theory that could provide useful points of departure for framing future, action-oriented research in this field. Each of these focusses at the three different levels articulated above and each draws on the nascent, but rapidly emerging, critical social science-based literature on energy and development [30]. They are:

1. National innovation systems theory and recent fusion, in low-income country contexts, with socio-technical transitions thinking.
2. Social practice theory.
3. Global political economy perspectives.

Nevertheless, as will become clear, several tensions and areas of disagreement exist between proponents of these different theoretical perspectives. Bearing in mind this paper's overarching concern with the transformative nature of PAYG in relation to achieving sustainable development, this raises a final question that future research in this field will need to grapple with, namely:

> How might the relative strengths and weaknesses of, and tensions between, the three areas of theory operationalized below contribute to a more comprehensive theoretical perspective on the potential transformative role of technology and innovation in low-income country contexts?

Below we suggest that a fourth, more overarching conceptual perspective, wedded to the importance of interdisciplinary co-constructed knowledge production, might be of use, namely:

4.　　Leach et al.'s [12] Pathways Approach.

This latter area of theory focusses on the dynamics of, and myriad potential pathways towards, sustainability, with an emphasis on the level of inclusivity and social justice attributed to alternative pathways. Adopting a pragmatic, epistemic position, and drawing on existing scholarship that utilises the Pathways Approach, we argue that each of the first three theoretical perspectives above can contribute, in different ways (casting analytical emphasis on different dimensions), to analysis of access to new technologies and innovation for sustainability transformations. But, importantly, this must build on emerging work that operationalizes these theoretical approaches within low-income contexts—contexts which represent stark differences to what Furlong [33] refers to as the "modern infrastructure ideal" that characterizes the empirical realities in the Global North from whence the first three theoretical perspectives above emerged.

In the sub-sections below, we explore the relevance of each of these areas of theory in turn.

*5.1. National Contexts for Fostering Innovation, Including Technology Development and Uptake*

Innovation systems thinking (e.g., [34]) provides a useful point of departure for thinking at national, more systemic levels about the dynamics of technology, innovation and development; the same level at which many mainstream commentators, as characterised in Section 1 above, tend to focus when discussing the transformative potential of new, PAYG financing and delivery models. The innovation systems literature can be broadly characterised as emphasising the systemic contexts within which the uptake of new technologies and the capacity for innovation has been, and can be, nurtured. Essentially, innovation systems refer to the network of actors (e.g., firms, universities, research institutes, government departments, and NGOs) within which technology development, transfer, and uptake occurs, the strength and nature of the relationships between them, and the institutional environment within which they operate [35]. Empirical analysis in the innovation studies literature has demonstrated that national differences in the nature, speed, and extent of technological change can be explained by a systemic understanding of the context within which technological change is facilitated, with innovation systems providing the context within which all processes of technology development, transfer, and uptake occur [35]. Ergo, it is these systemic contexts that policy and research aimed at delivering the SDGs ought to focus.

In many low-income countries, innovation systems are weak, fragmented, or non-existent, particularly concerning new, sustainable technologies. Research could, therefore, usefully focus on understanding the state of existing innovation systems around PAYG in different countries or regions. Questions around what resources different actors lack, or, indeed, what actors different countries lack, and the nature and strength of existing and potential relationships between relevant actors, could be answered. This would provide a fertile basis for directing and maximising the impact of resource investment aimed at building new, or strengthening existing, parts of innovation systems that have relevance to PAYG business models, including broader concerns with existing national policy frameworks and governance. Multiple actors would then be able to better target their investments and interventions, including national governments, NGOs, the private sector, donors, and inter-governmental actors (the latter two of which stand to make huge investments in technology and innovation around sustainability transformations in the near future, as they seek to deliver against the SDGs and international climate change commitments).

But innovation systems theory focusses at the level of firms, technologies, sectors, and national and regional economies—technology users and politics are relatively absent, or at least under emphasised [36]. Based on a detailed historical deconstruction of the success of solar PV in Kenya, Ockwell and Byrne [13] demonstrate the potential value of theoretical fusion between innovation systems perspectives and insights from socio-technical transitions thinking (e.g., [37,38]). The latter casts analytical attention on the ways technology and innovation co-evolve with social practices and broader social institutions, creating dominant socio-technical regimes and path dependency that new niches of sustainable technology struggle to compete with and influence in new, sustainable directions.

Importantly, as several authors have now shown in relation to energy access (e.g., [8,16,39–42]), these socio-technical regimes can look different in remote areas of low-income countries to those studied in the Global North; the latter being the empirical context within which the majority of the socio-technical transitions literature has emerged to date (as analysed, for example, in a recent special issue, see [43,44]). Nevertheless, as demonstrated by the aforementioned recent contributions, these low-income contexts are still subject to similar dynamics, where everyday practices and powerful economic and political interests align with dominant socio-technical regimes (e.g., in the supply of kerosene for cooking and lighting), or with potentially unsustainable alternatives (e.g., expanding grid-connected, coal-fired electricity supply), implying continued utility for socio-technical transitions perspectives within low-income contexts.

It is through the fusion between the component foci of these two bodies of theory that Ockwell and Byrne [13] argue we can move forward towards a theoretical perspective that can accommodate a more systemic understanding of technological change, innovation, and development, whilst simultaneously attending to the socio-technical nature of such change, and the powerful path-dependency of existing socio-technical regimes of energy consumption (and, in the case of PAYG business models, energy finance—see Rolffs et al. [8]). Based on this premise, Ockwell and Byrne [13] proffer a new, hybrid perspective on how to foster the uptake of new, sustainable technologies based on "socio-technical innovation system building" (something they have taken forward to inform a new policy perspective in the international climate finance field—see Ockwell and Byrne [35]). Whilst acknowledging the utility of Ockwell and Byrne's [13] synthesised perspective, and the detailed, historical empirical basis upon which they developed it, we would, however, argue that their analysis is left wanting in two key areas. Firstly, they fail to properly attend to social practices at the level of the lived, daily realities of poor and marginalised women and men. Secondly, whilst politics is, to some extent, acknowledged in the idea of socio-technical regimes, their analysis is, by their own admission (both in their 2017 monograph and a recent paper, see [16]), left wanting in its attention to politics and, in particular, the powerful political economic dynamics at play.

## 5.2. The Daily Lives of Poor and Marginalised Women and Men

Prominent social practice theorists have acknowledged socio-technical transitions theory for its explicit theory of social change (something other relevant areas of theory are arguably lacking—see Shove [45]). Nevertheless, social practice theory is arguably much more explicit than socio-technical transitions theory in emphasising the lived realities of the people whom it is assumed will benefit from access to new, sustainable technologies. In the context of low-income countries, it is these lived realities of poor people that arguably must be kept centre field in any research that seeks to contribute to achieving the SDGs. As Shove and Walker [46] convincingly argue, what is remiss in much energy policy research is explicit attention to the central question "what is energy for?" Poor women and men do not want PAYG business models. Nor do they want solar panels or access to the grid. They want, or need, things like being able read at night (and hence need light), or to communicate with others (hence needing to charge mobile phones), or to keep food fresh (hence needing refrigeration).

A social practice perspective allows us to push back against the technological determinism that traditionally characterises analysis of technology and innovation within the development literature [47]. Attention is focussed instead on the energy services poor people want or need (e.g., light, heating, cooling, and communications) and, fundamentally, the social practices these services facilitate (e.g., studying or pursuing economic activities after dark, sending money electronically, or communicating with family or business associates). This, in turn, provides the potential for critical interrogation of the extent to which PAYG business models are, or are not, meeting the self-constructed needs and aspirations of poor people, as well as how the nature of social practices might be changing (or not—for better or for worse) as a result.

Emphasising self-constructed needs and aspirations also focusses our attention on issues of social discrimination, most obviously including inter- and intra-household gender relations [48,49], but

also issues like income inequality, race, caste, etc., that might intersect in different ways with PAYG business models. For example, one PAYG business director interviewed during this research had noticed a difference between whether the man or the woman in a household ended up with agency over how electricity is used, based on whether the PAYG payment method was via pre-paid scratch card or mobile-enabled bank payments. It is not surprising, then, that several contemporary scholars researching the intersection between gender dynamics and energy access have utilised social practice approaches [41,50]. This suggests further utility for deepening social practice-based research on the intersection between PAYG and the lived realities of poor women and men in relation to gender relations and beyond (see, for example, emerging work by Cross and colleagues [51,52]).

*5.3. Global Value Chains and Political Economies*

The questions arising from the stakeholder consultation above emphasise how any analysis of PAYG business models and sustainable development would be remiss without explicit consideration of global political economy dynamics. This requires specific attention to who wins, who loses, how and why from PAYG business models globally, particularly as to where economic benefits (whether financial or technological capacity building—both of high significance to long-term economic development in low-income countries) are being accumulated. It takes us to what Lasswell [53] argued to be the root of any political analysis: in his words, "who gets what, when, how".

Whilst acknowledging the arguments of proponents of socio-technical transitions theory, in particular [54,55] that, if done properly, attention to regime and landscape levels of analysis already allows socio-technical transitions scholarship to engage with politics, we nevertheless argue that there is more work to be done on political economy considerations in particular, such that it takes us beyond the boundaries of what socio-technical transitions scholarship has facilitated to date. Indeed, several contemporary scholars have usefully demonstrated how a political economy focus can provide more depth and insight on the specific dimensions, or levels, on which socio-technical transitions scholars tend to focus. This includes work on the political economy of climate change and energy in Kenya, where Newell and Phillips [56] have demonstrated how political economy perspectives can add depth to understanding what transitions scholars would refer to as the "landscape" level of analysis. Byrne et al. [16] have also sought to elaborate on how political economy dynamics shape the building of socio-technical "niches" and their potential to influence socio-technical "regimes". Furthermore, authors looking across the piste of SDG-relevant innovation and development questions have increasingly begun to emphasise the critical importance of politics and political economy in defining and shaping sustainability transformations [14,15].

Bringing these theoretical considerations into the context of the questions and concerns that arose from our stakeholder consultation, stakeholders emphasised that many PAYG businesses are headquartered in Europe. They also raised questions as to the extent to which indigenous companies and innovators are involved in these companies, or could be involved in the future (e.g., via explicit collaboration between indigenous innovators and international PAYG businesses, or via potential future innovations around open-source PAYG software platforms). This was seen to raise fundamental questions as to the extent to which low-income countries were able to benefit in the long-term from the presence and activities of PAYG companies, and the extent to which PAYG businesses facilitate value accumulation in the countries within which they operate, with "value" understood broadly as including financial capital, technological capabilities, and broader indigenous capacity building. Or, is this another case of capital accumulation in the Global North? Such concerns also emphasise the potential utility of global value chain analysis (see [57]) in facilitating empirical analysis that is able to explicitly analyse where value is being accumulated via PAYG businesses. This also allows us to acknowledge and intersect with critical accounts of financialisation and accumulation in the context of ideas of transformations (e.g., [58,59]).

### 5.4. Overarching Conceptual Frameworks, Pragmatic, Interdisciplinary Thinking, and Transformations Towards Sustainability: The Potential Utility of the Pathways Approach

As we have argued, the three areas of theory above provide analytical purchase for future research focussed on answering the questions that arose from our stakeholder consultation, and they do so in ways that are potentially complementary—each attending to specific levels and types of concern where the others are weak. Nevertheless, as we acknowledged at the outset, there also exist key areas of tension between these theories. For example, the analytical fusion between innovation systems and socio-technical transitions thinking attempted by Ockwell and Byrne [13] would likely be viewed as unacceptable to some proponents of each theoretical perspective. The idea of researchers wedded to social practice theory and those wedded to innovation systems theory accepting one another's ontological and epistemic positions as potentially complementary seems more unlikely still. That said, in the case of political economy theory, examples do exist of scholars who have successfully acknowledged the value of innovation capabilities (a core tenet of innovation systems thinking) in their analysis (see [60]). And, as demonstrated in a recent special issue (see [30]), it seems to be increasingly common in the contemporary, critical social science literature on energy access to find examples of scholars using social practice, innovation systems, or political economy theory (the latter also including the work by Newell and Phillips [56] and Byrne et al. [16] mentioned in Section 5.3 above) to drill down in more detail on specific aspects of the multi-level perspective [38] that has been so influential in socio-technical transitions thinking.

Whilst not wishing to belittle in any way the epistemic and ontological concerns that protagonists of these different theoretical positions might raise, we would argue that the rapid transformations demanded to deliver against the SDGs simultaneously demands a level of pragmatism on behalf of the research community. This is in line with calls for inter-disciplinary research that increasingly characterise SDG-focussed funding programmes (such as the UK's controversial Global Challenges Research Fund), as well as international scientific panels like the Intergovernmental Panel on Climate Change. Without wishing to disappear down a rabbit hole on the long-standing debate on the virtues of interdisciplinarity, its drawbacks and the difficulty of achieving it in practice, we end this paper by making a fundamental point pertaining to plurality and diversity. Through this we point towards a body of (what might be described as) 'meta-level' thinking and theorising that has potential to convene interdisciplinary research, focussed on transformations towards sustainability, that preserves a critical perspective (with an explicit eye on social justice and environmental sustainability), whilst accommodating a plurality of different research voices from different disciplines and theoretical perspectives.

With the emergence of the new, critical social science voice in the energy access literature has come empirically grounded critiques of the previously dominant two-dimensional understanding of the problem as being simply one of needing new technology and finance to pay for this. This early framing of the problem resulted in the simultaneous domination of research in this field by the disciplines of engineering and economics [13,61]. Seeing energy access as a two-dimensional problem ignores multiple other dimensions of the problem, including socio-cultural and political dimensions, as well as more systemic perspectives on innovation and the path dependency between social and technical dimensions of socio-technical change, as captured by the various theoretical perspectives reviewed above. But care needs to be taken that proponents of these latter perspectives do not fall into the same trap that many of them have argued energy access research was plagued by in the first place. Each of the critical social science theories that have more recently engaged with the energy access problem emphasise different aspects of the issue. It should, therefore, not be too great a leap to argue that there may be value in drawing on insights from across the piste of what these different perspectives offer. Simultaneously, whilst seeking to move towards more multi-dimensional, sophisticated understandings of the dynamic interplay between technology, innovation and sustainable development, critical social science scholars ought not dismiss the fundamental role of engineering and economics, alongside socio-cultural and politically focussed approaches. This arguably entails a need

to work towards perspectives that can draw from multiple disciplines, supporting a more enlightened understanding of the context-specific, values-based and deeply political nature [62] of sustainability transformations [63]. Such sentiments are well captured in recent work by Stirling [17,63–65], although it could be argued that Stirling's pushback against, for example, socio-technical transitions and socio-ecological systems scholars can itself be characterised as denying the plurality in knowledge production we argue for here (although, of course, Stirling's critique is in fact levelled squarely at the ways in which socio-technical transitions scholars have had a tendency to adopt an overly managerial, techno-deterministic conception of change, which is insufficient to address the socio-cultural and political nature of socially just transformations, thus risking achieving the exact opposite—this therefore places Stirling's work very much in line with the argument that we are attempting to articulate in the current paper).

In addition, throughout much of the literature on sustainability transformations, we observe a commitment to fostering the co-production of knowledge with stakeholders in ways that recognise the plurality of different voices and perspectives on a problem and the value and importance of fostering such diversity in knowledge production. Knowledge co-production is also now explicitly demanded (nominally at least) by many research funders. It seems illogical, therefore, not to be able to apply similar principles of plurality and diversity to the theoretical perspectives that underpin research; why co-produce knowledge with diverse stakeholders if we are not willing to respect the knowledge produced by researchers with different epistemic and ontological perspectives than ourselves?

The Pathways Approach (developed by Leach et al. [12]) is arguably one theoretical framework that has been able to accommodate different disciplinary voices, whilst simultaneously keeping centre field concerns with the politics of knowledge production and the ways in which this might work for, or against, marginalised people and the environment. It is designed to facilitate exactly the kind of theoretically grounded, critical, interdisciplinary analysis that this paper advocates. In simple terms, it casts aside the idea of a single, accurate and normatively 'good' pathway or route to sustainable development, and emphasises the need to remain open to multiple alternative development pathways that countries and communities might pursue. This is particularly vital in the context of the complex, interrelated challenges resulting from the need to address poverty whilst simultaneously dealing with other (often competing) priorities such as climate change, environmental integrity, job creation, economic growth, and social justice. Most fundamentally, the Pathways Approach recognises that who you are shapes how you 'frame'—or understand—a problem or opportunity, and that—alongside powerful interests and technological trajectories—these understandings have a tendency to focus on specific development pathways favoured by powerful groups to the neglect of alternative perspectives. Or they might simply represent the received wisdom [66] of donors or government agencies, failing to appreciate the realities of a problem from the perspective of poor households or national firms. For instance, a poor household, a solar home systems vendor, a member of parliament, a multilateral development bank, and a multinational PAYG-based business might all frame the benefits, costs, and risks of PAYG (and the services it provides access to) in different ways. Those various framings will lead to different narratives being told about PAYG's role in development and different choices being made about the value of PAYG, including where, to whom, and via what specific variance of the different business models that are now proliferating such values can be leveraged. These considerations apply equally whether considering PAYG business models for delivering solar lighting in rural villages, or models for brokering international deals with multinational companies for building new, large-scale programmes to scale up the use of PAYG for access to different technologies and their attendant services in different contexts. At all levels, critical questions need to be asked about the distribution of benefits—who gains, who loses, and how can this be changed to better deliver against the self-defined development needs of poor and marginalised people and poor countries more broadly?

Importantly, in light of our core argument in this paper, the Pathways Approach also emphasises, and has been demonstrated to enable, the use of interdisciplinary research to facilitate such analysis. Drawing on theory and ideas from across a diverse range of disciplines in both the natural and

social sciences, the Pathways Approach has been applied across a wide range of different research domains, including: health and disease [67–69]; food and agriculture [70–73]; pastoralism [74]; energy and climate change [13]; technology and innovation [75,76]; urbanisation [77,78]; water and sanitation [79,80]; gender equality [81]; and many more besides. This body of work provides support for the potential utility of the Pathways Approach as an overarching framework with which to use different theoretical approaches to examine the kinds of interconnected issues that arose from the stakeholder consultation reported above. Our attention is thus drawn to the dynamics, motivations and outcomes of the various types of PAYG business model (e.g., what assumptions underpin the design of these models? Who are they designed to benefit and how, and to what extent do these assumed benefits play out in practice, particularly over longer time periods? Which models target the poorest households, and what can policy makers/donors do to support these models? Have the successes and failures of PAYG to date been the result of technical, social, political or economic issues, or a combination of these?). The end point being the support of a broad church of critical research that is specifically geared towards providing the empirical and conceptual basis for understanding how PAYG business models are, or are not, supporting social just, environmentally sustainable outcomes and how their contribution to such goals, as epitomised by the SDGs, might better be supported in the future.

## 6. Conclusions

To conclude, there has been much hype around the potential of PAYG business models to affect a step change in access to energy (and other SDG-relevant services) for poor and marginalised women and men. Given the rapid rate at which these business models are spreading and evolving, and the fascinating story of technological innovation in mobile technologies that (on the face of it) sits behind the PAYG story, such hype is arguably understandable. This is especially so given the need for rapid, transformative change that delivering against the SDGs and other global sustainability goals (e.g., those of the Paris Climate Accord) demands. But, as the results of the stakeholder consultation above demonstrate, it is early days for PAYG business models in terms of our understanding of the extent to which they are delivering, or could deliver, meaningful sustainability transformations (defined by a commitment to social justice and environmental sustainability). As we have seen, the published and grey literature on PAYG to date is at best patchy and far from affording any comprehensive, longitudinal, and theoretically grounded picture of how PAYG is delivering, or might deliver, against myriad concerns central to achieving sustainability transformations.

In this paper we have presented three areas of theory (socio-technical innovation system-building, social practice, and global political economy) that we argue have potential to support critical, action-oriented research at the three different levels of concern that arose from our stakeholder consultation (national contexts for fostering innovation, the daily lives of poor and marginalised women and men, global political economies, and value accumulation) and the related groups of stakeholder questions with which these three fields of theory are able to engage. Furthermore, we have attempted to make the case for epistemic pluralism in bringing to bear these different theoretical perspectives, perspectives that are often in tension with one another, under the broader, unifying roof of Leach et al.'s [12] Pathways Approach. Through this, we argue, future analysis of PAYG business models will have potential to retain a critical focus on the variegated benefits that accrue to different actors (users, international or indigenous businesses, national economies, governments, women, men, etc.), whilst retaining a commitment to use such critical research to inform meaningful change in policy and practice.

In these ways, this paper has sought to contribute to emerging debates on sustainability transformations, focussing in particular on the new phenomenon of PAYG business models. In particular, the analysis above contributes to the existing literature on PAYG by summarising a range of outstanding questions that stakeholders perceive to be of relevance to understanding the existing and potential contribution of PAYG to sustainability transformations. It has also sought

to provide the theoretical bases through which future research might usefully be framed in order to maximise this potential contribution. Transformations towards sustainability and achieving the SDGs demands urgent, far-reaching, interdisciplinary research and action. This urgency must be met with respect, diversity, and a commitment to pluralism and the co-construction of knowledge (both between knowledge producers and knowledge users, as much as between knowledge producers themselves). Without this, research, policy and practice risk falling prey to the same monolithic, two-dimensional thinking that has plagued the energy access field—and work on technology, innovation and development more broadly—to date. Two-dimensional perspectives are not fit for purpose. It is, we would argue, time for change, seasoned with a heavy dose of pragmatism.

**Author Contributions:** This article is partly based on earlier conceptual work by Byrne and Ockwell and partly on new empirical work by Atela, Mbeva and Chengo. Ockwell contributed to the latter research conceptually and in planning data collection. The data collection formed part of a larger programme of international research managed by Ely and the paper benefited significantly from interactions with international partners in that broader programme of research. Additional data was collected by Ockwell on the basis of contacts provided by, and scoping work conducted with Kasprowicz. The text of this article was written by Ockwell, on the basis of discussion and reflection between all of the authors. All authors contributed to reading, commenting on and editing earlier drafts of the paper.

**Funding:** The authors gratefully acknowledge funding from the ESRC [grant number ES/I021620/1] and the Transformations to Sustainability programme, which is coordinated by the International Social Science Council (ISSC) and funded by the Swedish International Development Cooperation Agency (Sida), and implemented in partnership with the National Research Foundation of South Africa (Grant Number ISSC2015-TKN150224114426).

**Data Statement:** New data presented in this paper was collected under the ISSC grant acknowledged above and is reported in two project reports which are available open access at https://steps-centre.org/project/pathways-network/. A number of additional interviews were also conducted under the ISSC grant on the understanding of them being anonymous and off the record due to commercial sensitivities. The ESRC grant paid for earlier conceptual work upon which this paper is based.

**Acknowledgments:** The authors gratefully acknowledge the ESRC [grant number ES/I021620/1] for financial support, as well as the Transformations to Sustainability programme, which is coordinated by the International Social Science Council (ISSC) and funded by the Swedish International Development Cooperation Agency (Sida), and implemented in partnership with the National Research Foundation of South Africa (Grant Number ISSC2015-TKN150224114426).

**Conflicts of Interest:** The authors declare no conflict of interest.

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
