# Peer review of "Can Pay-As-You-Go, Digitally Enabled Business Models Support Sustainability Transformations in Developing Countries? Outstanding Questions and a Theoretical Basis for Future Research"

_sustainability, doi:10.3390/su11072105_

Round 1

Reviewer 1 Report

This is a very interesting manuscript which addresses different dimensions of Pay-As-You-Go (PAYG) models for enabling better provision of basic services (with a focus on energy, water and agricultural services) in developing countries. The authors weave into their analysis interesting theoretical lenses that could strengthen understanding of PAYG models, including the literature on socio-technical and innovation studies, social practice and political economy theories. The authors highlight the Pathways approach as a way to bring into the analysis questions of social justice, sustainability and plurality of perspectives. 

I generally share their concern that social justice, politics and power and sustainability are especially inadequately addressed in PAYG literature. The paper was easy to read and very few if any spelling errors were noted. 

Although I do see scope for publication of this manuscript, I also have some suggestions for further reflection, improvements and clarifications. 

abstract

line 15: explain use of metamorphosing. Metamorphosing implies change but also something deeper like transformation which is also used in the paper. Does the choice of the two words have different implications for the argument and analysis? 

introduction

line 120: The paper articulates 'critical' questions. I think that it would be useful to be more explicit about the point of critique. Are these critical questions relevant for those already well versed with the theories highlighted or is it perhaps that you want to reach out to a community that perhaps 'needs to know' about these unaddressed questions and are working with completely different models and approaches? 

Line 121: The paper draws on a 'stakeholder consultation'. I think that this current phrasing is a bit generic and merits some more explanation. What is the background and motivation for this study in the first place? Why were these stakeholders consulted in the first place?  Did you use certain criteria to select these stakeholders? How did the stakeholders benefit in tangible ways by the transformation labs? 

Section on literature review 

Its a bit strange that the literature review starts with a more methodological description of the review process. Even though there is an entire methodology section later on where I would expect to see the rationale for the review. Could it be that the methodology section should come first before the review? 

Lines 180: The number of papers is indeed surprisingly small. I wonder, where any particular search criteria and-or keywords used as part of the literature search that could help widen the search? Consideration of inclusion-exclusion criteria? Are there any uncertainties or limitations associated with the way the authors approached the literature review? 

Section on results-questions

I think the way the questions are presented here is interesting but it could be useful to consider adding some new material that would strengthen this section. For example, linking the questions to a specific case study illustration? Or perhaps bringing in material from the interviews from the stakeholder consultations? How do the stakeholders talk about these questions and how do these findings relate to their livelihoods, professional routines etc.? 

Section on discussion

Reading this section gave me the impression that the different theoretical perspectives could be a bit better balanced. In a sense it is more clear how the Pathways approach may help us to arrive at a more inclusive model of PAYG but then why is the description of the other theoretical frameworks necessary? In a sense is it to show simply that Pathways is informed by these other perspectives or is it that each perspective (including pathways) has its own unique contributions to theorising better PAYG? There are examples within the different literature studies highlighted that emphasise inclusivity, plurality of perspectives and co-creation. This relates to the point about epistemic pragmatism.Do the authors perceive only one model of epistemic pragmatism (perhaps the pathways model?) or several different models? 

- Line 547: Subsection titles are too long. A minor suggestion would be to use same title phrasing as in lines 516-519? 

Reviewer 2 Report

I very much enjoyed reading this paper and learned a lot from the approach and findings presented here. The identification of critical questions that can guide the sustainability of PAYG business models is clearly directly relevant to the journal's readership and the identification of appropriate theoretical frameworks to address each will clearly guide readers interested in tackling these.

My only comment regarding the content of the paper itself is whether the clarity of the interactions between the four perspectives could be aided by some kind of diagram. Throughout the text, mention is given to the ways in which these perspectives can be both complimentary and contradicting. Such a diagram could tie together this commentary, offering a big picture view and helping to tie together future work that takes on any of these diverse approaches by highlighting the particular areas where there is overlap between the frameworks, likewise with conflict. This is not an essential revision, but I feel it could be of value, especially to those who are quickly reading through the paper without the time to fully delve into each section of rich detail.

The quality of English in this paper is far better than anything I have ever written, but I did spot one minor issue:

p3 l146 'Dis-benefits' sounds very awkward. Could 'drawbacks' be a better fit?
